# Entanglement-Assisted Quantum Codes from Cyclic Codes

**DOI:** 10.3390/e25010037

**Published:** 2022-12-24

**Authors:** Francisco Revson F. Pereira, Stefano Mancini

**Affiliations:** 1IQM Quantum Computers, Nymphenburger Str. 86, 80636 Munich, Germany; 2School of Science and Technology, University of Camerino, I-62032 Camerino, Italy; 3Istituto Nazionale di Fisica Nucleare, Sezione di Perugia, I-06123 Perugia, Italy

**Keywords:** quantum codes, Reed–Solomon codes, BCH codes, maximal distance separable, maximal entanglement

## Abstract

Entanglement-assisted quantum-error-correcting (EAQEC) codes are quantum codes which use entanglement as a resource. These codes can provide better error correction than the (entanglement unassisted) codes derived from the traditional stabilizer formalism. In this paper, we provide a general method to construct EAQEC codes from cyclic codes. Afterwards, the method is applied to Reed–Solomon codes, BCH codes, and general cyclic codes. We use the Euclidean and Hermitian construction of EAQEC codes. Three families have been created: two families of EAQEC codes are maximal distance separable (MDS), and one is almost MDS or almost near MDS. The comparison of the codes in this paper is mostly based on the quantum Singleton bound.

## 1. Introduction

Practical implementations of most quantum communication schemes and quantum computers will only be possible if such systems incorporate quantum-error-correcting codes. Quantum error correcting codes restore quantum states from corrupted by unwanted noisy action. One of the most known and used methods to create quantum codes from classical block codes is the CSS method [1]. Unfortunately, it requires (Euclidean or Hermitian) duality containing to one of the classical codes used. One way to overcome this constraint is via entanglement shared beforehand by the communicating parties. It is possible to show that such entanglement-assisted construction also improves the error-correction capability of quantum codes. These codes are called entanglement-assisted quantum error-correcting (EAQEC) codes. The first proposals of EAQEC codes were presented by Bowen [2] and Fattal et al. [3]. Then, Brun et al. [4] have developed an entanglement-assisted stabilizer formalism for these codes, which was recently generalized by Galindo et al. [5].

This formalism has created a method to construct EAQEC codes from classical block codes, which has lead to the construction of several families of EAQEC codes [6,7,8,9,10,11,12,13]. The majority of them utilize constacyclic codes [7,9,10,14] or negacyclic codes [8,9] as the classical counterpart. However, only a few of them have used cyclic codes and described the parameters of the quantum code constructed via the defining set of cyclic code. This can lead to a straightforward relation between the parameters of the classical and quantum codes and a method to create MDS EAQEC code. Li et al. used BCH codes to construct EAQEC codes via decomposing the defining set of the BCH code used [15]. Lu and Li constructed EAQEC codes from primitive quaternary BCH codes [16]. Recently, Lu et al. [9], using not cyclic but constacyclic MDS codes as the classical counterpart, proposed four families of MDS EAQEC codes.

Deriving EAQEC codes with different parameters provides a tool for reliable communication through quantum channels. In the quantum communication framework, sender and receiver are physically separated, which makes impossible the use joint unitary transformations. However, one can use other resources in order to maximize the code rate within the constraint of high minimum distance, such as pre-shared entanglement. Although in a specified communication scenario one may aim at deriving a high rate code with a low error probability for the block code, due to the broad approach followed in this paper (where we do not design codes for a particular type of quantum channel), we shall use the minimum distance as performance measure for EAQEC codes.

The main goal of this paper is to describe any cyclic code, such as Reed–Solomon and BCH codes, under the same framework via defining set description, and to show, using two classical codes from one of these families, how to construct EAQEC codes from them. We use Euclidean and Hermitian methods to construct EAQEC codes. As it will be shown, EAQEC codes from Reed–Solomon codes are MDS codes, and the ones from BCH codes are new in two senses. The first one is that there is no work in the literature with the same parameters. The second one is that we usie two BCH codes to derive the EAQEC code, which gives more freedom in the choice of parameters. Two more families of EAQEC codes are devised using the Hermitian construction. One of these families can generate codes which are almost MDS or almost near MDS; i.e., the Singleton defect for these codes is equal to one or two units. The last family created is maximally entangled and has a length proportional to a high power of the cardinality of the field. This family, when extending the block length, could approach the EA quantum hashing bound similarly to what happens to turbo codes in reference [17]. In fact, it was shown by Lai et al. [18] that maximal-entanglement EAQEC turbo codes get close to the EA quantum hashing bound. (By the EA quantum hashing bound is intended the quantum communication rate given by $1-\frac{1}{2}\left[p\left(\log_{2}3-\log_{2}p\right)-\left(1-p\right)\log_{2}\left(1-p\right)\right]$ for a depolarizing channel characterized by depolarizing parameter *p* [17].) Lastly, we would like to highlight that the description given in this paper gives a more direct relation between cyclic codes and the entanglement-assisted quantum codes constructed from them. Such a relation can be extended to constacyclic and negacyclic codes with a few adjustments.

The paper is organized as follows. In Section 2, we review Reed–Solomon and BCH codes and describe their parameters via a defining set. Additionally, we show construction methods of EAQEC codes from classical codes. Using these methods for cyclic classical codes, new EAQEC codes are constructed in Section 3. In Section 4, a comparison of these codes is presented via the quantum Singleton bound. In particular, we show families of MDS and almost MDS EAQEC codes. We also create a family of EAQEC codes which could approach the EA quantum hashing bound [17,18,19]. Lastly, the conclusion is presented in Section 5.

### Notation

Throughout this paper, *p* denotes a prime number and q≠2 is a power of *p*. Let $F_{q}$ be the finite field with *q* elements. A linear code *C* with parameters ${\left[n,k,d\right]}_{q}$ is a *k*-dimensional subspace of $F_{q}^{n}$ with minimum distance *d*. For cyclic codes, Z(C) denotes the defining set, and g(x) is the generator polynomial. Lastly, an ${\left[\left[n,k,d;c\right]\right]}_{q}$ quantum code is a $q^{k}$-dimensional subspace of $C^{q^{n}}$ with minimum distance *d* that utilizes *c* pre-shared entangled pairs.

## 2. Preliminaries

In this section, we review some ideas related to linear complementary dual (LCD) codes, cyclic codes, and entanglement-assisted quantum codes. Before giving a description of LCD codes, we need to define the Euclidean and Hermitian dual of a linear code.

**Definition** **1.**
*Let C be a linear code over $F_{q}$ with length n. The (Euclidean) dual of C is defined as*

(1)
$$C^{⊥}=\left\{\;x\in F_{q}^{n}\;|\;x\cdot c=0\;for\;all\;c\in C\right\}.$$

*If the finite field has cardinality equal to $q^{2}$, an even power of a prime, then we can define the Hermitian dual of C. This dual code is defined by*

(2)
$$C^{{⊥}_{h}}=\left\{\;x\in F_{q^{2}}^{n}\;|\;x\cdot c^{q}=0\;for\;all\;c\in C\;\right\},$$

*where $c^{q}=\left(c_{1}^{q},\ldots ,c_{n}^{q}\right)$ for $c\in F_{q^{2}}^{n}$.*


These types of dual codes can be used to derive quantum codes from the stabilizer formalism [1]. The requirement in this formalism is to the classical code to be self-dual; i.e., $C\subseteq C^{⊥}$ or $C\subseteq C^{{⊥}_{H}}$. However, there is a different relationship between a code and its (Euclidean or Hermitian) dual that can be interesting for constructing an EAQEC. This relation is complementary duality and is defined in the following.

**Definition** **2.**
*The hull of a linear code C is given by $hull\left(C\right)=C^{⊥}\cap C$. The code is called linear complementary dual (LCD) code if the hull is trivial; i.e, hull(C)={0}. Similarly, it is defined by $hull_{H}\left(C\right)=C^{{⊥}_{h}}\cap C$ and the idea of a Hermitian LCD code.*


Now, we can define cyclic codes and some properties that can be used to extract the parameters of the quantum code constructed from them.

### 2.1. Cyclic Codes

A linear code *C* with parameters ${\left[n,k,d\right]}_{q}$ is called cyclic if for any codeword $(c_{0},c_{1},\ldots ,$$c_{n-1})\;\in C$ implies $(c_{n-1},c_{0},c_{1},\ldots ,c_{n-2})\in C$. By defining a map from $F_{q}^{n}$ to $F_{q}\left[x\right]/\left(x^{n}-1\right)$, which takes $c=\left(c_{0},c_{1},\ldots ,c_{n-1}\right)\in F_{q}^{n}$ to $c\left(x\right)=c_{0}+c_{1}x+\cdots +c_{n-1}x^{n-1}\in F_{q}\left[x\right]/\left(x^{n}-1\right)$, we can see that a linear code *C* is cyclic if and only if it corresponds to an ideal of the ring $F_{q}\left[x\right]/\left(x^{n}-1\right)$. Since any ideal in $F_{q}\left[x\right]/\left(x^{n}-1\right)$ is principal, any cyclic code *C* is generated by a polynomial $g\left(x\right)|\left(x^{n}-1\right)$, which is called a generator polynomial. This polynomial is monic, and has the smallest degree among all the generators of *C*.

A characterization of the parameters of a cyclic code can be given from the generator polynomial and its defining set. For the description of this set, consider the following: Let $m=ord_{n}\left(q\right)$, α be a generator of the multiplicative group $F_{q^{m}}^{*}$, and assume $\beta =\alpha^{\frac{q^{m}-1}{n}}$; i.e., β is a primitive *n*-th root of unity. Then, the defining set of *C*, which is denoted by Z(C), is defined as $Z\left(C\right)=\{i\in Z_{n}:c\left(\beta^{i}\right)=0\;\mathrm{for}\;\mathrm{all}\;c\left(x\right)\in C\}$.

BCH and Reed–Solomon codes are particular cases of cyclic codes, where the generator polynomial has some additional properties. See Definitions 3 and 5.

**Definition** **3.**
*Let b≥0, δ≥1, and $\alpha \in F_{q^{m}}$, where $m=ord_{n}\left(q\right)$. A cyclic code C of length n over $F_{q}$ is a BCH code with designed distance δ if*

$$g\left(x\right)=lcm\{m_{b}\left(x\right),m_{b+1}\left(x\right),\ldots ,m_{b+\delta -2}\left(x\right)\}$$

*where $m_{i}\left(x\right)$ is the minimal polynomial of $\alpha^{i}$ over $F_{q}$. If $n=q^{m}-1$, then the BCH code is called primitive, and if b=1, it is called narrow-sense.*


Before relating the parameters of an BCH code with the defining set, we need to introduce the idea of the cyclotomic coset. It comes from the observation that the minimal polynomial $m_{i}\left(x\right)$ of $\alpha^{i}$ can be the minimal polynomial of other powers of α. The reason for this is that α belongs to an extension of $F_{q}$ while the polynomial $m_{i}\left(x\right)\in F_{q}\left[x\right]$. The set of all zeros of $m_{i}\left(x\right)$ in the field $F_{q^{m}}$ is given by the cyclotomic coset of *i*. Thus, the defining set of a BCH code *C* is the union of the cyclotomic cosets of b,b+1,…,b+δ−2. The following definition describes this set.

**Definition** **4.**
*The q-ary cyclotomic coset modn containing an element i is defined by*

(3)
$$C_{i}=\left\{i,iq,iq^{2},iq^{3},\ldots ,iq^{m_{i}-1}\right\},$$

*where $m_{i}$ is the smallest positive integer such that $iq^{m_{i}}\equiv i\;\mathrm{mod}\;n$.*


For the parameters of a BCH code, it is shown that the dimension is equal to n−|Z(C)| and the minimal distance of *C* is at least δ [20]. Thus, we can see that important properties of an BCH codes can be obtained from the defining set. The same characterization happens with Euclidean or Hermitian dual cyclic code. Propositions 1 and 2 focus on this.

**Proposition** **1**([20], Proposition 4.3.8)**.**
*Let C be a linear code of length n and defining set Z(C). Then, the defining set of $C^{⊥}$ is given by*
$$Z\left(C^{⊥}\right)=Z_{n}\backslash \left\{-i|i\in Z\left(C\right)\right\}$$
*For BCH codes, the generator polynomial is given by the lcm of the minimal polynomials over $F_{q}$ of the elements $\alpha^{j}$ such that $j\in Z(C^{⊥})$.*

**Proposition** **2.**
*Let C be a cyclic code over $F_{q^{2}}$ with defining set Z(C). Then,*

$$Z\left(C^{{⊥}_{h}}\right)=Z_{n}\backslash \left\{-i|i\in qZ\left(C\right)\right\}.$$



**Proof.** Let $c\in F_{q^{2}}^{n}$ be a codeword of *C*. By expressing $c^{q}$ as a polynomial, we have that $c^{\left(q\right)}\left(x\right)=c_{0}^{q}+c_{1}^{q}x+\cdots +c_{n-1}^{q}x^{n-1}$. Thus, $i\in Z_{n}$ belongs to $Z(C^{q})$ if and only if
$$\begin{array}{ccc} c^{\left(q\right)}\left(\alpha^{i}\right)=0 & \Leftrightarrow & c_{0}^{q}+c_{1}^{q}\alpha^{i}+\cdots +c_{n-1}^{q}\alpha^{i(n-1)}=0\\ & \Leftrightarrow & {\left(c_{0}^{q}+c_{1}^{q}\alpha^{i}+\cdots +c_{n-1}^{q}\alpha^{i(n-1)}\right)}^{q}=0\\ & \Leftrightarrow & c_{0}+c_{1}\alpha^{iq}+\cdots +c_{n-1}\alpha^{iq(n-1)}=0\\ & \Leftrightarrow & iq\in Z(C). \end{array}$$
This shows that $Z\left(C^{q}\right)=qZ\left(C\right)$. Since $C^{{⊥}_{h}}={\left(C^{q}\right)}^{⊥}$, we have from Proposition 1 that $Z\left(C^{{⊥}_{h}}\right)=Z_{n}\backslash \left\{-i|i\in qZ\left(C\right)\right\}$. □

The other class of cyclic codes used in this paper, Reed–Solomon codes, can be viewed as a subclass of BCH codes. Thus, a similar characterization in terms of defining set can be given; see Definition 5 and Corollary 1. One property of such codes that makes them important is that they are maximal distance separable (MDS) codes; i.e., fixing the length and the dimension, they have the maximal minimal distance possible. As shown in Section 3, using such codes to construct EAQEC codes will result in MDS quantum codes.

**Definition** **5.**
*Let b≥0, n=q−1, and 1≤k≤n. A cyclic code $RS_{k}\left(n,b\right)$ of length n over $F_{q}$ is a Reed–Solomon code with minimal distance n−k+1 if*

$$g\left(x\right)=\left(x-\alpha^{b}\right)\left(x-\alpha^{b+1}\right)\cdots \left(x-\alpha^{b+n-k-1}\right),$$

*where α is a primitive element of $F_{q}$.*


A particular application of Proposition 1 to Reed–Solomon codes is described in Corollary 1, where the parameters and defining set of an Euclidean dual of a Reed–Solomon are derived.

**Corollary** **1.**
*Let $RS_{k}\left(n,b\right)$ be a Reed–Solomon code. Then, its Euclidean dual can be described as*

$$RS_{k}{\left(n,b\right)}^{⊥}=RS_{n-k}\left(n,n-b+1\right)$$

*In particular, the defining set of $RS_{k}{\left(n,b\right)}^{⊥}$ is given by $Z(RS_{k}{\left(n,b\right)}^{⊥})=\left\{n-b+1,n-b+2,\ldots ,n-b+k\right\}$.*


As will be shown in the next subsection, the amount of entanglement in a EAQEC code is computed from the dimension of the intersection between the two codes. Thus, the last proposition of this subsection addresses the subject.

**Proposition** **3**([21], Exercise 239, Chapter 4)**.**
*Let $C_{1}$ and $C_{2}$ be cyclic codes with defining sets $Z(C_{1})$ and $Z(C_{2})$, respectively. Then, the defining set of $C_{1}\cap C_{2}$ is given by $Z\left(C_{1}\right)\cup Z\left(C_{2}\right)$. In particular, $\dim\left(C_{1}\cap C_{2}\right)=n-\left|Z\left(C_{1}\right)\cup Z\left(C_{2}\right)\right|$.*

### 2.2. Entanglement-Assisted Quantum Codes

**Definition** **6.**
*A quantum code Q is called an ${\left[\left[n,k,d;c\right]\right]}_{q}$ entanglement-assisted quantum error-correcting (EAQEC) code if it encodes k logical qudits into n physical qudits using c copies of maximally entangled states and can correct ⌊(d−1)/2⌋ quantum errors. A EAQEC code is said to have maximal entanglement when c=n−k.*


Formulating a stabilizer paradigm for EAQEC codes gives a way to use classical codes to construct this quantum codes [22]. In particular, we have the next two procedures by Galindo et al. [5].

**Proposition** **4**([5], Theorem 4)**.**
*Let $C_{1}$ and $C_{2}$ be two linear codes over $F_{q}$ with parameters ${\left[n,k_{1},d_{1}\right]}_{q}$ and ${\left[n,k_{2},d_{2}\right]}_{q}$; and parity check matrices $H_{1}$ and $H_{2}$, respectively. Then, there is a EAQEC code with parameters ${\left[\left[n,k_{1}+k_{2}-n+c,d;c\right]\right]}_{q}$, where $d=\min\{d_{H}\left(C_{1}\backslash \left(C_{1}\cap C_{2}^{⊥}\right)\right),d_{H}\left(C_{2}\backslash \left(C_{1}^{⊥}\cap C_{2}\right)\right)\}$, with $d_{H}$ as the minimum Hamming weight of the vectors in the set, and*
(4)$$c=\mathrm{rank}\left(H_{1}H_{2}^{T}\right)=\dim C_{1}^{⊥}-\dim\left(C_{1}^{⊥}\cap C_{2}\right)$$
*is the number of required maximally entangled states.*

**Proposition** **5**([5], Proposition 3 and Corollary 1). *Let C be a linear codes over $F_{q^{2}}$ with parameters ${\left[n,k,d\right]}_{q}$, H be a parity check matrix for C, and $H^{*}$ be the q-th power of the transpose matrix of H. Then, there is a EAQEC code with parameters ${\left[\left[n,2k-n+c,d^{\prime };c\right]\right]}_{q}$, where $d^{\prime }=d_{H}\left(C\backslash \left(C\cap C^{{⊥}_{h}}\right)\right)$, with $d_{H}$ as the minimum Hamming weight of the vectors in the set, and*
(5)$$c=\mathrm{rank}\left(HH^{*}\right)=\dim C^{{⊥}_{h}}-\dim\left(C^{{⊥}_{h}}\cap C\right)$$
*is the number of required maximally entangled states.*

A measurement of goodness for a EAQEC code is the quantum Singleton bound (QSB). Let ${\left[\left[n,k,d;c\right]\right]}_{q}$ be a EAQEC code. Then, the QSB is given by
(6)$$d\leq \lfloor \frac{n-k+c}{2}\rfloor +1.$$
The difference between the QSB and *d* is called a quantum Singleton defect. When the quantum Singleton defect is equal to zero (resp. one), the code is called the maximum distance separable quantum code (resp. almost maximum distance separable quantum code), and it is denoted the MDS quantum code (resp. almost MDS quantum code).

## 3. New Entanglement-Assisted Quantum-Error-Correcting Cyclic Codes

In this section is shown the construction of EAQEC codes from the cyclic codes. We are going to make use of Euclidean and Hermitian constructions, which will give codes with different parameters when compared over the same field.

### 3.1. Euclidean Construction

A straightforward application of cyclic codes to the Proposition 4 via defining set description can produce some interesting results. See Theorem 1 and Corollary 2.

**Theorem** **1.**
*Let $C_{1}$ and $C_{2}$ be two cyclic codes with parameters ${\left[n,k_{1},d_{1}\right]}_{q}$ and ${\left[n,k_{2},d_{2}\right]}_{q}$, respectively. Then, there is an EAQEC code with parameters $[[n,k_{1}-|Z\left(C_{1}^{⊥}\right)\cap Z\left(C_{2}\right)|,\min\left\{d_{1},d_{2}\right\};n-k_{2}-|Z\left(C_{1}^{⊥}\right)\cap Z\left(C_{2}\right){\left|]\right]}_{q}$.*


**Proof.** From Proposition 3, we have that $\dim\left(C_{1}^{⊥}\cap C_{2}\right)=n-|Z\left(C_{1}^{⊥}\right)\cup Z\left(C_{2}\right)|\;=n-|Z\left(C_{2}\right)|-|Z\left(C_{1}^{⊥}\right)|+|Z\left(C_{1}^{⊥}\right)\cap Z\left(C_{2}\right)|\;=k_{2}-k_{1}+\left|Z\left(C_{1}^{⊥}\right)\cap Z\left(C_{2}\right)\right|$. Thus, the amount of entanglement used in an EAQEC code constructed from these two cyclic codes can be computed from Proposition 4, which is $c=n-k_{2}-\left|Z\left(C_{1}^{⊥}\right)\cap Z\left(C_{2}\right)\right|$. By substituting this value of *c* in the parameters of the EAQEC code in Proposition 4, we obtain an $[[n,k_{1}-|Z\left(C_{1}^{⊥}\right)\cap Z\left(C_{2}\right)|,\min\left\{d_{1},d_{2}\right\};n-k_{2}-|Z\left(C_{1}^{⊥}\right)\cap Z\left(C_{2}\right){\left|]\right]}_{q}$ EAQEC code. □

**Corollary** **2.**
*Let C be a LCD cyclic code with parameters ${\left[n,k,d\right]}_{q}$. Then, there is a maximal entanglement EAQEC code with parameters ${\left[\left[n,k,d;n-k\right]\right]}_{q}$. In particular, if C is MDS, so is the EAQEC code derived from it.*


**Proof.** Let $C_{1}=C_{2}=C$ in Theorem 1. Since *C* is LCD, $|Z\left(C_{1}^{⊥}\right)\cap Z\left(C_{2}\right)|\;=0$. From Theorem 1, we have that there is an EAQEC code with parameters ${\left[\left[n,k,d;n-k\right]\right]}_{q}$. □

**Theorem** **2.**
*Let $C_{1}=RS_{k_{1}}\left(n,b_{1}\right)$ and $C_{2}=RS_{k_{2}}\left(n,b_{2}\right)$ be two Reed–Solomon codes over $F_{q}$ with $0\leq b_{1}\leq k_{1}$, $b_{2}\geq 0$, and $b_{1}+b_{2}\leq k_{2}+1$. Then, we have two possible cases:*
*1.* 
*For $k_{1}-b_{1}\geq b_{2}$, there is an EAQEC code with parameters*

$${\left[\left[n,b_{1}+b_{2}-1,n-\min\left\{k_{1},k_{2}\right\}+1;n+b_{1}+b_{2}-k_{1}-k_{2}-1\right]\right]}_{q};$$

*2.* 
*For $k_{1}-b_{1}<b_{2}$, there is an EAQEC code with parameters*

$${\left[\left[n,k_{1},n-\min\left\{k_{1},k_{2}\right\}+1;n-k_{2}\right]\right]}_{q}.$$




**Proof.** From Corollary 1, we have that $Z\left(C_{1}^{⊥}\right)=\left\{n-b_{1}+1,n-b_{1}+2,\ldots ,n-b_{1}+k_{1}\right\}$. First of all, notice that the restriction $b_{1}+b_{2}\leq k_{2}+1$ implies that the first element in the defining set of $Z(C_{1}^{⊥})$ comes after the last element in $Z(C_{2})$. Since $0\leq b_{1}\leq k_{1}$, we have that $n-b_{1}+k_{1}\geq n$, which implies that the defining set for $C_{1}^{⊥}$ equals to $Z\left(C_{1}^{⊥}\right)=\left\{n-b_{1}+1,n-b+2,\ldots ,n-1,0,1,\ldots ,k_{1}-b_{1}\right\}$. Thus, $Z(C_{1}^{⊥})$ intersects $Z(C_{2})$ if and only if $k_{1}-b_{1}\geq b_{2}$. In the case that it does, the intersection is equals to $Z\left(C_{1}^{⊥}\right)\cap Z\left(C_{2}\right)=k_{1}-\left(b_{1}+b_{2}\right)+1$. The missing claims are obtained using these results in Theorem 1. □

**Corollary** **3.**
*Let $C=RS_{k}\left(n,b\right)$ be a Reed–Solomon code over $F_{q}$ with 0<b≤(k+1)/2 and 0<k<n≤q. Then, there is an MDS EAQEC code with parameters ${\left[\left[n,2b-1,n-k+1;n+2b-2k-1\right]\right]}_{q}$. In particular, for b=(k+1)/2, there is a maximal entanglement MDS EAQEC code.*


**Proof.** Let $C_{1}=C_{2}=RS_{k}\left(n,b\right)$ in Theorem 2. Assuming 0≤b<(k+1)/2, we have that the classical codes fall in the first case of Theorem 2; and for b=(k+1)/2, we are in the second case of Theorem 2. Thus, substituting the values of $k_{1},k_{2}$ and $b_{1},b_{2}$ by *k* and *b*, respectively; the result follows. □

In a similar way, we can use BCH codes to construct EAQEC codes. The advantage in using BCH codes is that the length of the code is not bounded by the cardinality of the finite field used. However, creating classical or quantum codes from BCH codes which are MDS is a difficult task. Our proposal to have BCH codes as the classical counterpart in this paper is to show how to use two BCH codes to construct EAQEC codes. In addition, it is also constructed maximal entanglement EAQEC codes. In order to do this, we show suitable properties concerning some cyclotomic cosets for $n=q^{2}-1$.

**Lemma** **1.**
*Let $n=q^{2}-1$ with q>2. Then, the q-ary coset $C_{0}$ has one element, and $C_{i}=\left\{i,iq\right\}$ for any 1≤i≤q−1.*


**Proof.** The first claim is trivial. For the second one, notice $iq^{2}\equiv i\;\mathrm{mod}\;\left(q^{2}-1\right)$. Thus, the only elements in $C_{i}$ are *i* and iq, for 1≤i≤q−1. □

From Lemma 1, we can construct EAQEC codes with length $n=q^{2}-1$. See Theorem 3.

**Theorem** **3.**
*Let $n=q^{2}-1$ with q>2. Assume a,b are integers such that 0≤a≤q−1 and 1≤b≤q. Then, there is an EAQEC code with parameters*

*${\left[\left[n,2\left(q-b\right)-1,b+1;2\left(q-a-1\right)\right]\right]}_{q}$, if a≥q−b and b<q;*

*${\left[\left[n,2a+1,b+1;2b-\left\lfloor \frac{b}{q}\right\rfloor \right]\right]}_{q}$, if a<q−b.*



**Proof.** First of all, assume that $C_{1}^{⊥}$ has a defining set given by $Z\left(C_{1}^{⊥}\right)=\cup_{i=0}^{a}C_{i}$, and the defining set of $C_{2}$ is equal to $Z\left(C_{2}\right)=\cup_{i=1}^{b}C_{q-i}$. From Lemma 1, we have that $|Z(C_{1}^{⊥})|=2a+1$ and $|Z\left(C_{2}\right)|\;=2b-\left\lfloor \frac{b}{q}\right\rfloor$. Thus, the dimensions of $C_{1}$ and $C_{2}$ are equal to $k_{1}=\left|Z\left(C_{1}^{⊥}\right)\right|=2a+1$ and $k_{2}=n-\left|Z\left(C_{2}\right)\right|=n-2b+\left\lfloor \frac{b}{q}\right\rfloor$, respectively. To compute $|Z\left(C_{1}^{⊥}\right)\cap Z\left(C_{2}\right)|$, we have to consider two cases. If a≥q−b, then we have that $Z\left(C_{1}^{⊥}\right)\cap Z\left(C_{2}\right)=\cup_{i=q-b}^{a}C_{i}$, which has cardinality given by $|Z\left(C_{1}^{⊥}\right)\cap Z\left(C_{2}\right)|\;=2\left(a-\left(q-b\right)+1\right)-\left\lfloor \frac{b}{q}\right\rfloor$, because $|C_{0}|\;=1$. On the other hand, if a<q−b, then $|Z\left(C_{1}^{⊥}\right)\cap Z\left(C_{2}\right)|\;=0$. Lastly, since a,b≤q, $Z\left(C_{1}^{⊥}\right)=\cup_{i=0}^{a}C_{i}$, and $n=q^{2}-1$ with q>2, we can see that $d_{1}>d_{2}=b+1$. Now, using these results in Theorem 1, we have that there is a EAQEC code with parameters ${\left[\left[n,2\left(q-b\right)-1+\left\lfloor \frac{b}{q}\right\rfloor ,b+1;2\left(q-a-1\right)\right]\right]}_{q}$, if a≥q−b, or a EAQEC code with parameters ${\left[\left[n,2a+1,b+1;2b-\left\lfloor \frac{b}{q}\right\rfloor \right]\right]}_{q}$. □

### 3.2. Hermitian Construction

In the same way as before, it possible to use cyclic codes to construct EAQEC codes from the Hermitian construction method of Proposition 5. See the following theorem.

**Theorem** **4.**
*Let C be a cyclic code with parameters ${\left[n,k,d\right]}_{q^{2}}$. Then there is an EAQEC code with parameters $[[n,k-|Z\left(C^{{⊥}_{h}}\right)\cap Z\left(C\right)|,d;n-k-|Z\left(C^{{⊥}_{h}}\right)\cap Z\left(C\right){\left|]\right]}_{q}$.*


**Proof.** First of all, from Proposition 3 we have $\dim\left(C^{⊥}\cap C\right)=n-|Z\left(C^{⊥}\right)\cup Z\left(C\right)|\;=n-|Z\left(C\right)|-|Z\left(C^{{⊥}_{h}}\right)|+|Z\left(C^{{⊥}_{h}}\right)\cap Z\left(C\right)|\;=k-k+|Z\left(C^{{⊥}_{h}}\right)\cap Z\left(C\right)|\;=|Z\left(C^{{⊥}_{h}}\right)\cap Z\left(C\right)|$. Thus, $c=\dim\left(C^{{⊥}_{h}}\right)-\dim\left(C^{⊥}\cap C\right)=n-k-\left|Z\left(C^{{⊥}_{h}}\right)\cap Z\left(C\right)\right|$. Using a ${\left[n,k,d\right]}_{q^{2}}$ to construct a EAQEC codes via Proposition 5, we derive a code with parameters $[[n,k-|Z\left(C^{{⊥}_{h}}\right)\cap Z\left(C\right)|,d;n-k-|Z\left(C^{{⊥}_{h}}\right)\cap Z\left(C\right){\left|]\right]}_{q}$. □

**Corollary** **4.**
*Let C be an LCD cyclic code with parameters ${\left[n,k,d\right]}_{q^{2}}$. Then there is a maximal entanglement EAQEC code with parameters ${\left[\left[n,k,d;n-k\right]\right]}_{q}$.*


**Proof.** From the proof of Theorem 4, we have that $\dim\left(C^{{⊥}_{h}}\cap C\right)=\left|Z\left(C^{{⊥}_{h}}\right)\cap Z\left(C\right)\right|$. Since *C* is LCD, $|Z\left(C^{{⊥}_{h}}\right)\cap Z\left(C\right)|\;=0$, and the result follows from Theorem 4. □

Differently from the construction of EAQEC code via Euclidean dual cyclic code, the construction via Hermitian dual can be more delicate, even for Reed–Solomon codes. Even so, we are going to show that is possible to construct EAQEC codes from a particular case of Reed–Solomon codes and some cyclic codes.

**Theorem** **5.**
*Let q be a prime power and assume $C=RS_{k}\left(n,1\right)$ is a Reed–Solomon code over $F_{q^{2}}$ with $k=qt+r<q^{2}$, where t≥1 and 0≤r≤q−1, and $n=q^{2}$. Then we have the following:*

*If t≥q−r−1, then there exists an MDS EAQEC code with parameters*

$${\left[\left[q^{2},{\left(t+1\right)}^{2}-2\left(q-r\right)+1,q\left(q-t\right)-r+1;{\left(q-t-1\right)}^{2}+1\right]\right]}_{q}.$$


*If t<q−r−1, then there exists an MDS EAQEC code with parameters*

$$\left[\left[q^{2},t^{2}-1,q\left(q-t\right)-r+1;{\left(q-t\right)}^{2}-2r-1\right)\right]{]}_{q}.$$




**Proof.** Since $C=RS_{k}\left(n,0\right)$, we have that Z(C)={0,1,2,…,n−k−1}. From the proof of Theorem 2, we also have that $Z\left(C^{{⊥}_{h}}\right)=qZ\left(C^{⊥}\right)=\left\{q,2q,\ldots ,kq\right\}$. From $n=q^{2}$ and k=qt+r, we can rewrite these two defining sets as Z(C)={qi+j|0≤i≤q−t−2,0≤j≤q−1}∪{(q−t−1)q+j|0≤j≤q−r−2} and $Z\left(C^{{⊥}_{h}}\right)=\left\{qi+j|0\leq i\leq q-1,0\leq j\leq t-1\right\}\cup \left\{qi+t|0\leq i\leq r\right\}$. Using this description, we can compute $|Z\left(C\right)\cap Z\left(C^{{⊥}_{h}}\right)|$. To do so, we have to consider two cases separately, t≥q−r−1 and t<q−r−1. For the first case, the intersection is given by the following set $Z\left(C\right)\cap Z\left(C^{{⊥}_{h}}\right)=\left\{qi+j|0\leq i\leq q-t-2,0\leq j\leq t\right\}\cup \left\{\left(q-t-1\right)q+j|0\leq j\leq q-r-2\right\}$. Thus, $|Z\left(C\right)\cap Z\left(C^{{⊥}_{h}}\right)|\;=\left(q-t-1\right)\left(t+1\right)+q-r-1$. Similarly for the case t<q−r−1, we have $Z\left(C\right)\cap Z\left(C^{{⊥}_{h}}\right)=\left\{qi+j|0\leq i\leq q-t-1,0\leq j\leq t-1\right\}\cup \left\{qi+t|0\leq i\leq r\right\}$, which implies $|Z\left(C\right)\cap Z\left(C^{{⊥}_{h}}\right)|\;=\left(q-t\right)t+r+1$. Using these results and the fact that *C* has parameters ${\left[q^{2},k,q^{2}-k+1\right]}_{q^{2}}$, in Theorem 4, we have that there exists a EAQEC code with parameters
${\left[\left[q^{2},{\left(t+1\right)}^{2}-2\left(q-r\right)+1,q\left(q-t\right)-r+1;{\left(q-t-1\right)}^{2}+1\right]\right]}_{q}$, for t≥q−r−1; and$\left[\left[q^{2},t^{2}-1,q\left(q-t\right)-r+1;{\left(q-t\right)}^{2}-2r-1\right)\right]{]}_{q}$, for t<q−r−1.□

**Theorem** **6.**
*Let $n=q^{4}-1$ and q≥3, a prime power. There exists an EAQEC code with parameters ${\left[\left[n,n-4\left(a-1\right)-3,d\geq a+1;1\right]\right]}_{q}$, where $2\leq a\leq q^{2}-1$.*


**Proof.** Let $C_{a}$ be a cyclic code with defining set $Z\left(C_{a}\right)=C_{0}\cup C_{q^{2}+1}\cup \left(\cup_{i=2}^{a}C_{q^{2}+a}\right)$, for $2\leq a\leq q^{2}-1$. From Ref. [23], we have that $C_{q^{2}+1}=\left\{q^{2}+1\right\}$ and $C_{q^{2}+a}=\left\{q^{2}+a,1+aq^{2}\right\}$. It is trivial to show that $C_{0}=\left\{0\right\}$. From $-qZ\left(C_{a}\right)\cap Z\left(C_{a}\right)=C_{0}$ [23], we can see that $Z\left(C_{a}^{{⊥}_{h}}\right)\cap Z\left(C_{a}\right)=Z\left(C_{a}\right)\backslash C_{0}$. Hence, $|Z\left(C_{a}^{{⊥}_{h}}\right)\cap Z\left(C_{a}\right)|\;=2\left(a-1\right)+1$. From the assumption of the defining set, the dimension and minimal distance of the classical code are k=n−2(a−1)−2 and d≥a+1, respectively. Thus, using these quantities in Theorem 4, we have that there exists an EAQEC code with parameters ${\left[\left[n,n-4\left(a-1\right)-3,d\geq a+1;1\right]\right]}_{q}$. □

Two important comments can be made about Theorem 6. Comparing the bound given for the minimal distance and the Singleton bound for EAQEC codes, we see that the difference between these two values is equal to a−1. Thus, for lower values of *a* (such as a=2 or a=3), the EAQEC codes have a minimal distance, close to the optimum; e.g., if a=2 (or a=3), the family of EAQEC codes is almost MDS (or almost MDS). The second point is that the codes in Theorem 6 can be seen as a generalization of the result by Qian and Zhang [24].

In the following, we use LCD cyclic code to construct maximal entanglement EAQEC codes. The families obtained have an interesting range of possible parameters.

**Theorem** **7.**
*Let q be a prime power, m≥2, $2\leq \delta \leq q^{2\lceil \frac{m}{2}\rceil }+1$, and $\kappa =q^{2m}-2-2\left(\delta -1-\lfloor \frac{\delta -1}{q^{2}}\rfloor \right)m$. Then,*
*1.* 
*For m odd and 1≤u≤q−1, there is a maximal entanglement EAQEC code with parameters ${\left[\left[q^{2m}-1,k,d\geq \delta +1+\left\lfloor \frac{\delta -1}{q}\right\rfloor ;q^{2m}-1-k\right]\right]}_{q}$, where*

(7)
$$\begin{array}{c} k=\left\{\begin{array}{cc} \kappa , & if\;2\leq \delta \leq q^{m}-1;\\ \kappa +u^{2}m, & if\;uq^{m}\leq \delta \leq \left(u+1\right)\left(q^{m}-1\right);\\ \kappa +(u^{2}+2v+1)m, & if\;\delta =\left(u+1\right)\left(q^{m}-1\right)+v+1\;for\;0\leq v\leq u-1;\\ \kappa +q^{2}m, & if\;\delta =q^{m+1}\;or\;q^{m+1}+1. \end{array}\right. \end{array}$$

*2.* 
*For m even, there is an maximal entanglement EAQEC code with parameters*

(8)
$${\left[\left[q^{2m}-1,\kappa ,d\geq \delta +1+\left\lfloor \frac{\delta -1}{q}\right\rfloor ;2\left(\delta -1-\left\lfloor \frac{\delta -1}{q^{2}}\right\rfloor \right)m+1\right]\right]}_{q}.$$




**Proof.** From Li [25], we have that there are LCD cyclic codes with parameters ${\left[q^{2m}-1,k,\delta +1+\left\lfloor \frac{\delta -1}{q}\right\rfloor \right]}_{q^{2}}$, where *k* is the same as in Equations (Equation 7) and (Equation 8) for odd and even *m*, respectively. Thus, by applying this LCD code to Corollary 4, we obtain the mentioned codes. □

## 4. Code Examples

In Table 1, we present some MDS EAQEC codes obtained from Corollary 3 and Theorem 5. The codes in the first column are obtained from the Euclidean construction and the ones in the second column from the Hermitian construction. As can be seen, the latter one has a higher length within the same field. Thus, it can be used in applications where the underline quantum system has limited dimensions. On the other hand, the codes in the first column can have parameters that the ones from the Hermitian construction cannot. Thus, these two classes of EAQEC codes are suitable for specific applications.

The codes obtained from Corollary 3 and Theorem 5 are maximal entanglement EAQEC codes. We could use the dependency between the cardinality of the finite field and code parameters to derive new codes. In particular, this is not the case for the codes in Ref. [26], where the cardinality of the finite field must be two. Additionally, one cannot find in Ref. [9] codes similar to the ones on the left column of Table 1, since the codes in Ref. [9] request a number *c* of entangled pairs that can be only equal to one or two. For our codes with c=1 or 4, which can be used in a comparison with the codes in Ref. [9], we see that the codes ${\left[\left[4,3,2;1\right]\right]}_{4}$ and ${\left[\left[13,9,5;4\right]\right]}_{13}$ have parameters slightly worse than the codes ${\left[\left[5,4,2;1\right]\right]}_{7}$ and ${\left[\left[10,9,5;4\right]\right]}_{3}$, respectively. Lastly, if we do not take into consideration the cardinality of the field, we continue to see improvements in the code parameters. As an example, the code ${\left[[16,3,9;3]\right]}_{4}$ has a higher rate (ratio between code dimension and code length) than the similar minimum distance code ${\left[\left[31,10,10;21\right]\right]}_{4}$ given in Ref. [27].

One family of EAQEC codes derived from BCH codes has been constructed; see Theorem 3. Some examples of these EAQEC codes are shown in Table 2. As can be seen in Table 1 in Ref. [28] (and the reference there in), the EAQEC codes derived from Theorem 3 have new parameters when compared with EAQEC codes known in the literature. Thus, though not having good parameters as the ones in our Table 1 in terms of quantum Singleton defect, these codes are new. One advantage of our codes with respect to the ones known in the literature is that, since they are constructed from two BCH codes, we have more freedom in the choice of parameters. The family of codes presented in Table 2 could be used in environments with low amounts of resources, since we have more freedom in the code parameters. As an example, the codes in Table 2 are longer than the codes in Table 1 for the same cardinality of the finite field, making the codes in Table 2 more favorable to environments where increasing the size of individual systems is less costly than composing such systems. Looking at the examples of Table 2, we see that there is no counterpart for the codes with parameters ${\left[[63,7,5;8]\right]}_{8}$ and ${\left[[255,19,7;12]\right]}_{16}$ in Ref. [26]. However, we did not obtain an improvement in rate when comparing the remaining codes in Table 2 with the codes shown in Ref. [29].

The remaining EAQEC codes constructed in this paper are the ones derived from cyclic codes that are neither Reed–Solomon nor BCH codes. Two families of such codes were created, both of them using Hermitian construction. Some examples of parameters that can be obtained from these codes are presented in Table 3. Codes in the first column are almost MDS or almost MDS—i.e., the Singleton defect, which is when the difference between the quantum Singleton bound (QSB) presented in Equation (Equation 6) and the minimal distance of the code is equal to one or two units. Lastly, we display in the second column of Table 3 some codes from Theorem 7. All codes in Theorem 7 are maximal entanglement. Thus, this family, when extending the block length, could approach the EA quantum hashing bound similarly to what happens to turbo codes in Ref. [17]. Having length proportional to a high power of the cardinality of the field, it is expected to achieve low error probability using these codes.

To compare the codes shown in Table 2 and Table 3, we are going to use the concepts of ratio, given by k/n, and net ratio, given by (k−c)/n, where k,c, and *n* are the code dimension, the number of maximally entangled states, and code length, respectively. For the code ${\left[[80,50,10;30]\right]}_{3}$, we see significant improvements in rate and net rate when comparing with the codes ${\left[\left[73,36,10;37\right]\right]}_{4}$ and ${\left[\left[89,44,10;45\right]\right]}_{4}$ shown in Ref. [27]. A similar conclusion is obtained for the comparison between our ${\left[[255,237,7;18]\right]}_{4}$ and the code ${\left[\left[217,186,6;31\right]\right]}_{4}$ shown in Ref. [27]. Lastly, we also have codes with no counterpart in Ref. [27], such as ${\left[[80,73,3;1]\right]}_{3}$ and ${\left[[255,248,3;1]\right]}_{4}$, due to large discrepancy in code parameters.

## 5. Conclusions

This paper has been devoted to the use of cyclic codes in the construction of EAQEC codes. General construction methods of EAQEC codes from cyclic codes via defining sets have been presented, using both Euclidean and Hermitian duals of the classical codes. As an application of these methods, five families of EAQEC codes were created. Two of them were derived from Reed–Solomon codes, which resulted in MDS codes. An additional family of almost MDS or near almost MDS EAQEC codes was derived from general cyclic codes. One of the remaining family used BCH codes as the classical counterpart. The construction of this family of EAQEC code used two BCH codes, which provided more freedom in the parameters of the quantum code. Lastly, we conjecture that the family of constructed EAQEC codes can achieve the hashing bound when extending their length. This is supported by the fact that the codes derived have maximal entanglement. Investigations (mainly numerical) along this line are left for future work.

## Figures and Tables

**Table 1 entropy-25-00037-t001:** Some new MDS EAQEC codes from Reed–Solomon codes. The codes with a star ⋆ are maximal entanglement MDS EAQEC codes.

New EAQEC codes—Corollary 3	New EAQEC codes—Theorem 5
${\left[\left[n,2b-1,n-k+1;n+2b-2k-1\right]\right]}_{q}$	$\left[\left[q^{2},t^{2}-1,q\left(q-t\right)-r+1;{\left(q-t\right)}^{2}-2r-1\right)\right]{]}_{q}$
0<b≤(k+1)/2 and 0<k<n≤q	$qt+r<q^{2}$, where 1≤t<q−r−1 and 0≤r≤q−1
Examples	
$⋆{\left[[3,1,3;2]\right]}_{3}$	${\left[[16,3,9;3]\right]}_{4}$
$⋆{\left[[4,3,2;1]\right]}_{4}$	${\left[[64,35,17;3]\right]}_{8}$
$⋆{\left[[7,3,5;4]\right]}_{7}$	${\left[[64,15,31;11]\right]}_{8}$
$⋆{\left[[8,5,4;3]\right]}_{8}$	${\left[[256,196,33;3]\right]}_{16}$
$⋆{\left[[11,9,3;2]\right]}_{11}$	${\left[[256,120,78;18]\right]}_{16}$
$⋆{\left[[13,9,5;4]\right]}_{13}$	${\left[[1024,784,129;15]\right]}_{32}$
${\left[[16,13,3;2]\right]}_{16}$	${\left[[1024,624,220;38]\right]}_{32}$

**Table 2 entropy-25-00037-t002:** Some new EAQEC codes from BCH codes.

New EAQEC codes—Theorem 3
${\left[\left[q^{2}-1,2a+1,b+1;2b-\left\lfloor \frac{b}{q}\right\rfloor \right]\right]}_{q}$
1≤b≤q and 0≤a<q−b
Examples
${\left[[15,5,2;2]\right]}_{4}$
${\left[[48,9,3;4]\right]}_{7}$
${\left[[63,7,5;8]\right]}_{8}$
${\left[[255,19,7;12]\right]}_{16}$

**Table 3 entropy-25-00037-t003:** Some EAQEC codes from cyclic codes via Hermitian construction.

New EAQEC codes—Theorem 6	New EAQEC codes—Theorem 7
Examples	
${\left[[80,73,3;1]\right]}_{3}$	${\left[[80,42,14;38]\right]}_{3}$
${\left[[80,69,4;1]\right]}_{3}$	${\left[[80,50,10;30]\right]}_{3}$
${\left[[255,248,3;1]\right]}_{4}$	${\left[[255,193,20;62]\right]}_{4}$
${\left[[255,244,4;1]\right]}_{4}$	${\left[[255,237,7;18]\right]}_{4}$

## Data Availability

Not applicable.

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
