# Peer review of "Entanglement-Assisted Quantum Codes from Cyclic Codes"

_entropy, 2022, doi:10.3390/e25010037_

Round 1

Reviewer 1 Report

The authors propose a general method to construct EAQEC codes from cyclic codes. The authors applied the advocated method for creating new EAQEC codes to the classical Reed-Solomon, BCH, and general cyclic codes using the Euclidian and Hermitian dual codes constructions. The technical aspect of the paper looks solid. I only have some minor comments.

1. Page 1. "One way to overcome this constraint is via entanglement. It is possible to show that entanglement also improves the error-correction capability of quantum codes." Although it's already used prominently in the literature, I would suggest that the word "entanglement" in this context is replaced with "entanglement-assisted construction" just for clarity. The overall QEC constructions have already relied on entanglement, but what makes EAQEC codes special is pre-shared maximally entangled pairs are available beforehand.

2. The authors mention hashing bound on pages 2 and 9. I am not sure if the authors actually mean hashing bound or Hamming bound, because hashing bound is related to the one-shot capacity of qubits transmission over noisy quantum channels and Hamming bound is related to the code construction itself. Please clarify.

3. It would be nice to see the direct comparisons of the newly proposed codes with the ones from [12, 21, 22, 24] so the readers can immediately see the improvement of the parameters of the codes.

Overall, the paper looks solid mathematically, but some minor improvements can be included to improve the readability of the paper. Thank you very much and good luck.

Author Response

Thank you very much for your suggestions. We have considered all of your comments. Please, see detailed reply for each of them below:

1. We have made the change “entanglement” → “entanglementassisted
construction.” Please, see the first paragraph in Introduction.

2. In the points you highlighted, we meant “hashing bounds.” Please, see
C.-Y. Lai, T. A. Brun, and M. M. Wilde, “Duality in entanglement-assisted quantum error correction,” IEEE Transactions on Information Theory, vol. 59, no. 6, pp. 4020–4024, Jun. 2013
R. Li, L. Guo, and Z. Xu, “Entanglement-assisted quantum codes achieving the quantum Singleton
bound but violating the quantum Hamming bound,” Quantum Information & Computation, vol. 14, no. 13, pp. 1107–1116, Oct. 2014
The work by Lai, et al. shows that maximal-entanglement EAQEC codes are close to the hashing bound. Therefore, statements about hashing bound achieving codes follows from the constructed maximal-entanglement EAQEC codes presented in the paper. We have added the first reference to
our paper and keeping the reference by Li, et al. since it is one of the first works giving numerical examples of such codes. Please, see the new paragraph in Introduction.

3.  We added a comparison between our code parameters
and the codes in Refs. [12, 21, 22, 24]. Please, see Section Code Examples.

Reviewer 2 Report

The authors study the construction of entanglement-assisted quantum error correction codes from classical cyclic codes. The constructions are described in detail by proving every claim. I consider the article not to be a groundbreaking piece, but anyway interesting for the community, as well as scientifically sound.

However I consider that there are some important points that need to be addressed by the authors before it is publishable in Entropy:

- The authors present some code examples in section 4 and refer to other articles for comparison. I consider that such comparisons should be done in the paper since it is unclear which ones are better than the others. This would make the paper to be more fair in terms of the conclusions that the reader can get to determine which code is more interesting for some task.

- The authors claim in section 4 that "So, even not having good parameters as the ones in our Table 1, these codes are new", which I assume that their constructions are worse in terms of performance (in this case I guess that the distance they get for some k,n). I think that the authors should explain this more and then follow to discuss why they codes might be more interesting than others. They claim parameter flexibility, but this is a broad claim, meaningless if there is no context behind. They should put examples of when such flexibility is actually benefitial even if  performance is lost. In the present version of  the article, the authors just claim that their constructions are interesting just because they are new.

- In addition, they comment that their codes can achieve the hashing bound. They claim this in the introduction, in section 4 and in the conclusion. I think that this claim is vague in terms of the analysis they do.  First of all, the authors do not describe what do they refer to with hashing bound. The hashing bound refers to unassisted error correction codes, and their constructions are EA. Thus, such is not the quantity they should achieve. They should instead use the term entanglement-assisted hashing bound, which takes into account the entanglement consumption used in the quantum error correction task. They should explain such quantity correctly and justify why their codes can achieve such limit. Also, if the codes achieve the capacity they should be optimal (note that when maximal entanglement is used, the EA hashing bound is the capacity since there is no code degeneracy), so I then their codes would actually be better than others in the literature, thing that the authors claim they are not, leading to contradiction. I really consider they should study this further, if such claims need to be maintained. Also, they reference [34] for this, and I have not been able to access such document.

- Finally, I think that they should also give a little bit more of context regarding the use of these codes. They should discuss which quantum information tasks would they be useful for (i.e. communications, memories ...). Also which noise are they aimed to correct. I understand that the paper is more mathematically oriented, and I am aware of several works like this one, but I think that it would make their paper richer in terms of how other people of the field may use their constructions for other purposes. Specially, they actually comment that "Such property can help to adjust the EAQEC code to the framework where it will be used", but they do not specify any framework, such claim is just vague.

There are some other minor thigs such as the ordering of the references when cited through the text, but I consider that they are not critical. Thus, I think that the paper might be interesting for the reader, but to begin, the issues I raised should be addressed.

Author Response

Thank you very much for your suggestions. We have considered all of your comments. Please, see detailed reply for each of them below:

1. We added a comparison between our code parameters
and the codes in Refs. [12, 21, 22, 24]. Please, see Section Code Examples.

2. We have elaborated more on possible situations where codes with
more flexibility can be a suitable choice. Please, see Section 4.

3. The statements in our paper about hashing bound follow from the two papers below:
C.-Y. Lai, T. A. Brun, and M. M. Wilde, “Duality in entanglement-assisted quantum error correction,” IEEE Transactions on Information Theory, vol. 59, no. 6, pp. 4020–4024, Jun. 2013
R. Li, L. Guo, and Z. Xu, “Entanglement-assisted quantum codes achieving the quantum Singleton bound but violating the quantum Hamming bound,” Quantum Information & Computation, vol.
14, no. 13, pp. 1107–1116, Oct. 2014
In particular, the work by Lai, et al. shows that maximal-entanglement EAQEC codes are close to the hashing bound. Therefore, statements about hashing bound achieving codes follows from the constructed maximal-entanglement EAQEC codes presented in the paper. We have added the
first reference to our paper and an the above explanation to Introduction.

4. We have given some more context in Introduction.
Please, see the red marked paragraph there.

Round 2

Reviewer 1 Report

The authors have addressed all my comments satisfactorily. I have no further comments.

Author Response

We thank you for judging already our previous version satisfatory.

Reviewer 2 Report

Thank you for the changes made. I still think that adding the specific codes of references [12, 21, 22, 24] in the tables where the codes proposed by the authors would give clarity to the text, since in the present form, one has to go from the article to the references and find the specific codes they are comparing too. I think that is uncomfortable for the reader, but I do not think that it is a major issue.

Regarding my comments about the hashing bound, I still think that the authors are missing the point here. The authors refer to reference [17] by Lai et al. to justify their claim that their codes can achieve the hashing bound. I assume that they refer to footnote 2 in such article that states the following: "One might wonder why we are considering EAQEC codes that exploit the maximum amount of entanglement possible, given that noiseless entanglement could be expensive in practice. But there is good reason for doing so. The so-called father protocol is a random EA quantum code [23], [24], and it achieves the EA quantum capacity of a depolarizing channel (the EA hashing bound [12], [25]) by exploiting maximal entanglement. Furthermore, there is numerical evidence that maximal-entanglement turbo codes come within a few decibels of achieving the EA hashing bound [26]." However, note that they do not refer to the fact that every maximal EA code achieves the EA hashing bound, they state that it is known that the EA hashing bound (which turns out to be the channel capacity since the presence of maximal entanglement makes implies that there is no degeneracy) is achievable via random coding in the asymptotic regime. They actually specify that EA quantum turbo codes are few decibels from such limit, and those codes use long blocklengths (n=18000 or so), while from construction have a random structure. This is why those codes are near the hashing limit, specially due to blocklenths that are not asymptotical (this is impractical obviously), but are high enough to make the codes close to the limit. This obviously does not apply for the codes discussed in this work, since they restrict to low blocklengths. I guess that longer codes could be constructed, but I do not think that the claim "those codes would be capacity approaching codes" holds just from the fact that there exist codes that achieve capacity in the asymptotic regime. Moreover, the authors do not specify which EA hashing bound would their codes achieve, if they do. With this I mean that, the authors state that the capacities for depolarizing channels (which is what [17] refers to) or for other types of asymmetric quantum channels are different, and need other code structures to deal with such asymmetry. Actually they do not even specify that the hashing bound they are referring to is the entanglement-assisted hashing bound, since they use EA. The authors claim in the introduction that "where we do not design codes for a particular type of quantum channel", which is fine form the distance point of view, but when discussing a capacity approaching design, they should actually discuss the channel.

In addition to this, the authors claim that they construct maximal entanglement codes, which are defined as codes for which the entanglement consumption is c = n-k. However, inspecting Table 1, one finds codes such as [[16, 13, 3; 2]]16 on the left column or the ones in the right column for which such c=n-k does not hold. Thus, it seems that those codes are not maximal entangled codes. What's going on there? Also they state "one cannot find similar codes as the ones on the left column of Table 1 in Ref. [9], since the codes in Ref. [9] request more entanglement than ours", but what does it mean that they require more entanglement than their codes? Which codes of ref [9] are they comparing to? What do they mean that more entanglement is needed for the codes in ref [9]? are they speaking about the value of c or the rate of entanglement consumption?

I consider that all these things should be considered before the document is ready for publication.

Author Response

Thank you very much for your comments. We have carefully considered them. Please, see the detailed reply for each of them below.

1)  We have specifically mentioned EA quantum hashing bound whenever we talk about hashing bound, this was our point since the beginning. The
passages have also rephrased that one could achieve the EA quantum hashing bound using our maximal-entanglement codes with large code-length.

2) The codes presented in Table 1 are derived from Corollary
16 and Theorem 21. As it is in Corollary 16 and Theorem 21, one can obtain maximal-entanglement EAQEC codes if the some parameters are carefully chosen. So, we wanted to give examples of codes from Corollary 16 and Theorem 21 where the code parameters have some interesting
property, which may not be c = n − k. We rewrote Table 1 and highlighted with a star ⋆ the codes that are maximal-entanglement.

3) We apologize for the confusion. When we talk about entanglement, we mean entangled pairs c. We have changed the passage in the paper to clarify this confusion. Additionally, we explicitly point the codes we are comparing to.

We are confident that the present version is suitable for publication.

Round 3

Reviewer 2 Report

I think that the paper is now much clearer than before. I just want to comment the fact that even though I cannot disagree with the claim that their codes will achieve the EA hashing bound for the depolarizing channel when the blocklength is sufficiently large, I think that, in principle, that should not be true. For assuring so (at least numerically) they should study how the logical error probability of those "long" blocklength codes approach the limit. With this I just want to say that increasing the blocklength does not always assure that an specific family of codes will achieve the limit. From my point of view, and since the approach taken by the authors is to study the codes algebraically (which is completely valid) with the distance as a measure, I would not bother in discussing the fact that the limit is approachable by their codes. This comes from the fact that, their claim is just a conjecture until it is proven.